

# Recent results from LHCb for astroparticle physics

**Hans Dembinski⋆ on behalf of the LHCb collaboration**

TU Dortmund University, Dortmund, Germany

⋆ hans.dembinski@tu-dortmund.de

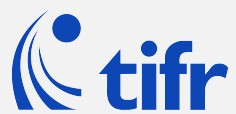

*21st International Symposium on Very High
Energy Cosmic Ray Interactions (ISVHECRI 2022)
Online, 23-28 May 2022*

## Abstract

**The LHCb experiment is a general-purpose forward spectrometer designed for the study of heavy flavour physics at the LHC. The acceptance in the pseudorapidity range $2 < \eta < 5$ with full tracking and particle identification capabilities down to very small transverse momentum make LHCb also ideal to study hadron production in the forward region. Measuring and modelling these processes is essential for the simulation of interactions of high-energy cosmic rays with matter, like Earth's atmosphere or the interstellar medium. We present recently published analyses from the LHCb collaboration relevant for this application.**


## 1 Introduction

The LHCb experiment [1, 2] is a single-arm spectrometer designed for the study of heavy flavour physics at the LHC. It is the only general purpose LHC experiment fully instrumented with tracking, particle identification, calorimeters, and a muon system in the pseudo-rapidity region $2 < \eta < 5$. These capabilities make LHCb also interesting for the study of the non-perturbative sector of quantum chromodynamics (QCD) in high-energy pp collisions and in heavy-ion collisions.

Because of the forward acceptance, LHCb provides the best constrains on (nuclear) parton density functions in the low-$x$ region [3], where $x$ is the momentum fraction of the nucleus. Studies of light hadron production in proton-proton and proton-lead collisions with LHCb help to solve the Muon Puzzle in cosmic-ray induced air showers [4], where a muon deficit is observed in simulated air showers in comparison with measurements [5–7]. The origin of this discrepancy should be visible at the LHC [8–12].

We present recent LHCb measurements on hadron production important for astroparticle physics. In several analyses, deviations from the QCD factorization framework [13] are observed in collisions with a high density of emitted particles, which seem to modify the hadron

composition in contradiction to the standard framework where fragmentation does not depend on environmental factors [14, 15]. Alternative fragmentation models which better fit these data are discussed in the literature; among them are the core-corona model [16, 17], fragmentation after string-string interactions [18–20], and models of direct quark coalescence [21].

## 2 Prompt $D^0$ production in proton-lead

Prompt $D^0$ production was studied in proton-lead at 8.16 TeV [22]. In comparison with measurements in proton-proton collisions, this probes so-called cold nuclear matter effects. These are initial state effects that can be encoded in the nuclear parton density functions (nPDFs), which are parameterizations that need to be fit to measurements. Due to the forward acceptance, $D^0$ production at LHCb probes nPDFs at the parton momentum fraction down to $x \sim 10^{-6}$, further than any other current experiment.

These results are important for astroparticle physics; directly, because $D^0$ decay is the main contribution to the high-energy prompt atmospheric neutrino flux seen as a background to astroneutrinos at neutrino observatories like IceCube [23–25]; and indirectly, since nPDFs enter any perturbative calculation of a hadron-nucleus production cross-section.

LHCb is able to discriminate between $D^0$ mesons produced directly in the primary collision (prompt) or via decay of $b$-hadrons (non-prompt), thanks to its high vertex resolution and the forward geometry. The measurement consists of the double differential production cross-section in transverse momentum, $p_T$ and rapidity, $y^*$, in the centre-of-mass system of the colliding nucleons, as well as the nuclear modification factor $R_{pPb}$ in these variables. The proton-proton reference cross-section was interpolated from earlier measurements at 5 and 13 TeV. A $D^0$ suppression is observed at forward rapidities (particles emitted at small angle to the proton), and an enhancement at backward rapidities (particles emitted at small angle to lead), which is generally in good agreement with predictions.

## 3 Prompt charged particles in proton-proton and proton-lead

Prompt charged particles were studied in two separate analyses; in proton-proton collisions at 13 TeV [26, 27], and at 5 TeV in proton-proton and proton-lead collisions [26, 27]. Both analyses report the double differential cross-sections in $p_T$ and pseudorapidity $\eta$; the latter also reports the nuclear modification factor. No interpolation of the proton-proton reference cross-section is necessary here thanks to the matching energies. Charged particles are largely pions, kaons, and protons, which also dominate the development of cosmic-ray induced air showers, therefore these measurements are important to tune and validate models, and have an impact on the Muon Puzzle. Understanding the nuclear effects in hadron production is important, since Earth's atmosphere is consists of nitrogen, oxygen, and argon.

The challenge for these analyses is to identify non-prompt contributions and to subtract them based on simulations, after adjusting the simulation prediction with control measurements. This can be done very precisely, however, both analyses reach total uncertainties down to about 2-3 %. The high precision is needed to sufficiently constrain the models used in astroparticle physics.

The proton-proton study at 13 TeV revealed deviations of model predictions from the measured cross-section of about 30 %, especially at high $p_T$. An asymmetry is observed the ratio of positively and negatively charged particles at $\eta > 4$ and $p_T > 3$ GeV/$c$ due to the excess of positive charge in the initial state. This feature is correctly reproduced by Pythia 8 [28], but not by other generators that were compared in the study.

In the study at 5 TeV, the nuclear modification factor is close to 1 at high $p_T$, but suppressed at forward rapidities for $p_T \lesssim 6\,\text{GeV}/c$. In the backward region, a significant enhancement is observed around $p_T \sim 3\,\text{GeV}/c$. Models based on QCD factorization match the forward measurements but not the backward measurements.

# 4   Neutral pion production in proton-lead collisions

Neutral pion production was measured in proton-lead collisions at 8.16 TeV [29]. These studies are complementary to those of charged particles, since they constrain the pion component in charged particle production and have different systematic uncertainties. Neutral pion production plays an important role for air shower development. They have a large impact on the depth of shower maximum and the total number of muons produced in the air shower.

Reconstructing neutral pions emitted in the forward region is challenging. The photons emitted in the decay have a small opening angle, which for large rapidities becomes smaller than the cell size of the LHCb electromagnetic calorimeter. If both photons end up in the same cell, the mass of the pion decay candidate cannot be reconstructed, but the mass is needed to separate real decays from background. A special reconstruction was therefore employed in this study. Only one photon was measured by the calorimeter, while the other was indirectly detected through its conversion into an electron-positron pair inside the detector material. This technique was very successful; clear peaks in the mass distributions were observed and the momentum resolution of the $\pi^0$ candidates was improved.

The double differential production cross-section and the nuclear modification factor are reported as a function of pseudorapidity and $p_T$. The proton-proton reference is interpolated from previous measurements at 5 and 13 TeV. The modification factor agrees well with that obtained with charged particles (see previous section) in the forward region, while the backward region the enhancement is smaller for neutral pions than for charged particles. Tensions are again observed between the data and to models based on QCD factorization.

# 5   $B_s^0/B^0$ ratio in proton-proton collisions

The $B_s^0$ and $B^0$ production cross-section ratio was measured in proton-proton collisions at 13 TeV [30]. The ratio is studied for the first time as a function of the number of charged tracks measured in the VELO, the central tracker of LHCb. The cross-section ratio is proportional to the ratio of probabilities for a $b$-quark to form a meson with an $s$-quark or a $d$-quark, and therefore sensitive to a potential strangeness enhancement in high-multiplicity events. Multiplicity-dependent strangeness enhancement was previously observed at mid-rapidity by ALICE [31], but not in the forward region. If this effect is also present in the forward region, it could be a key mechanism to resolve the Muon Puzzle in air showers.

Both $B$-mesons were reconstructed via the decay $B_s^0 \to J\Psi(\to e^+e^-)\pi^+\pi^-$. The systematic uncertainty of the measured ratio is strongly reduced thanks to several factors. The decay geometry and particle identification of pions allows one to measure the $B$ peaks in the mass distribution almost background-free. Since the decays are very similar, the various detector and selection efficiencies that enter the cross-section ratio cancel.

The $B$-meson ratio is shown in Fig. 1. The result presents $3.4\,\sigma$ evidence for an increase in the ratio as a function of the number of forward VELO tracks above the typical level observed in $e^+e^-$ collisions. This is the first evidence for multiplicity-dependent strangeness enhancement in the forward region. The positive slope is mainly seen at low transverse momentum, $p_T < 6\,\text{GeV}/c$, and vanishes at high $p_T$. It also vanishes if the ratio is measured as a function of

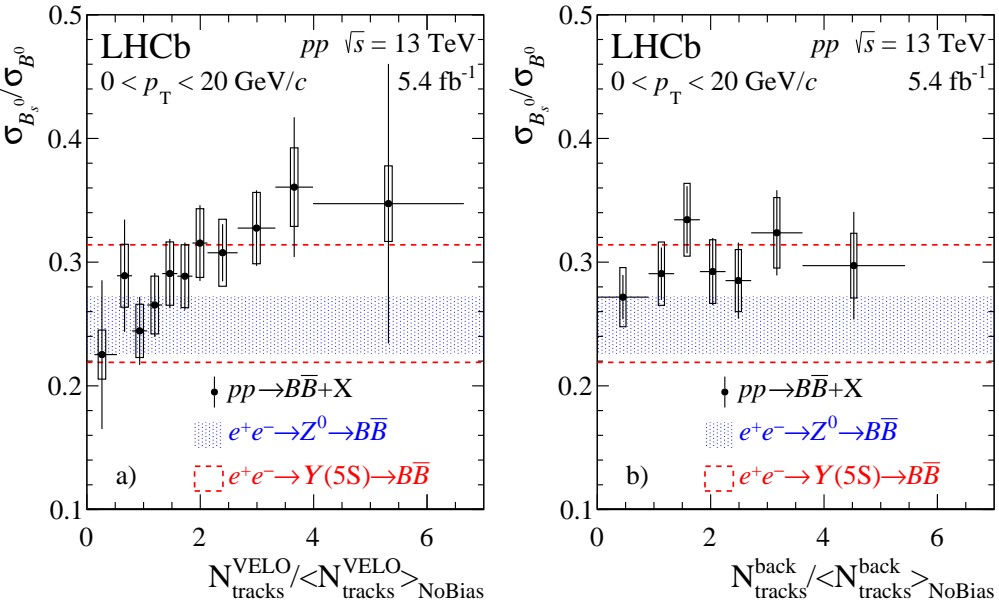

Figure 1: Ratio of cross-sections for $B_s^0$ and $B^0$ production as function of the normalized track multiplicity measured by the VELO (LHCb central tracker). Shown on the left-hand side is the number of VELO tracks which are roughly facing in the same direction as the $B$ mesons. On the right-hand side, the ratio is shown as a function of the VELO tracks which point into the opposite hemisphere. Error bars (boxes) indicate uncorrelated (fully correlated) uncertainties. Horizontal bands show values measured in $e^+e^-$ collisions for comparison. Figure reproduced from Ref. [30].

backward-facing VELO tracks. Since the correlation of the density of forward- and backward-facing tracks is small, this result may indicate that the local (in rapidity) environment around the produced $b$-quarks is important for the enhancement.

## 6 Non-prompt anti-protons from hyperon decays

The production of non-prompt anti-protons [32] from hyperon decays was studied in proton-helium collisions at $\sqrt{s_{NN}} = 110\,\text{GeV}/c$ by running LHCb in fixed-target mode. These collisions are studied by injecting small amounts of helium gas into the VELO, the central tracker of LHCb. The LHC proton beams then collide with this gas. The study is a follow-up on a previous measurement of prompt anti-proton production [33], where the uncertainty of the non-prompt contribution from hyperon decays was one of the largest contributions to the systematic uncertainty of that measurement. The anti-proton production cross-section is important for astroparticle physics, since the anti-proton/proton ratio in cosmic rays is used in searches for dark matter. Cosmic anti-protons produced in interactions of ordinary cosmic rays with the interstellar medium – which consists of about 9 % helium – are the main background for these searches. Furthermore, the cross-section is a tracer of baryogenesis in cosmic-ray interactions with matter, which has an impact on muon production in air showers.

The non-prompt anti-proton production is measured in two complementary ways. The first approach is to measure the decay $\bar{\Delta} \to \bar{p}\pi^+$ explicitly, which contributes about 70 % of the anti-protons from hyperon decays. This is done in the usual way. The other approach is to identify anti-protons from hyperon decays via their offset from the reconstructed primary

vertex, parametrized by the $\chi^2_{\text{IP}}$ variable. The distribution of this variable has three components, prompt anti-protons, anti-protons from hyperon decays, and anti-protons from particle interactions with the detector material. The three components are overlapping, therefore, the shapes need to be known well to measure the respective yields. This was ensured with detailed studies of the relevant components in simulations, and by adjusting the simulation with control measurements. The analysis results are reported as the ratio $R_{\bar{H}}$ of the non-prompt and prompt anti-proton production cross-sections. The observed ratio is larger than any model prediction, especially at low $p_{\text{T}}$.

## 7 Conclusion

The LHCb experiment offers unique opportunities for astroparticle physics in general, and the Muon Puzzle in air showers in particular. The forward acceptance of the tracking with particle identification is important, and the unique ability to study collisions of LHC beams with noble gasses. Several analyses by LHCb and the other LHC experiments show deviations from models based on QCD factorization. The most recent ones are listed in this proceeding. These measurements are important for the simulation of interactions of cosmic rays with matter at centre-of-mass energies beyond the LHC, where the particle densities nevertheless similar to proton-lead collisions at the LHC energy scale. If the effects seen at the LHC are indeed universal (depend only on particle density, but not on the collision system, or the centre-of-mass energy), it will be possible to safely extrapolate them to the higher collision energies that occur in the first interactions of cosmic rays with the atmosphere, which are beyond the reach of colliders.

## Acknowledgements

**Funding information** The author acknowledges funding by the Deutsche Forschungsgemeinschaft (DFG, German Research Foundation) – project no. 449728698.

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
