# Peer review of "Recent results from LHCb for astroparticle physics"

_SciPost Physics Proceedings, doi:SciPost Phys. Proc. 13, 016 (2023)_

## Round 1 · Referee Report · Anonymous · 2022-9-24

Report
The manuscript is already well-written and clear. There needs no major comment.
I would ask to correct typos and suggest a few minor comments for further improvement.
1) In Section 1, "the cora-corona model" should be "the core-corona model".
2) In Section 2, it would be good to give a description or definition of the nuclear medication factor, R_pPb.
3) In Section 3, root(s_NN) would be first introduced here for the proton-lead collisions at 8.16 TeV, since you use it later in Section 6.
4) In Section 5, it may be good to indicate the rapidity coverage of VELO if it is available. Also is it possible to clarify the tracks at VELO mainly come from non-diffractive collisions or diffractive collisions ?
Author: Hans Dembinski on 2022-10-04 [id 2871]
(in reply to Report 1 on 2022-09-24)Thank you for the review. I made edits to clarify points 1 to 4. Regarding the question whether VELO tracks mainly come from non-diffractive or diffractive collisions, I don't know for sure and I don't know where to look it up. At mid-rapidity, diffractive effects contribute about 25 % of the total cross-section, but I don't know how many particles, and I don't know how that changes when you further forward. The fraction of particles produced by diffractive events should increase with |Ī·|.
Anonymous on 2022-09-24 [id 2848]
The manuscript is already well-written and clear. There needs no major comment. I would ask to correct typos and suggest a few minor comments for further improvement.
1) In Section 1, "the cora-corona model" should be "the core-corona model".
2) In Section 2, it would be good to give a description or definition of the nuclear medication factor, R_pPb.
3) In Section 3, root(s_NN) would be first introduced here for the proton-lead collisions at 8.16 TeV, since you use it later in Section 6.
4) In Section 5, it may be good to indicate the rapidity coverage of VELO if it is available. Also is it possible to clarify the tracks at VELO mainly come from non-diffractive collisions or diffractive collisions ?

---

## Round 2 · Referee Report · Anonymous (Referee 2) · 2022-10-8

Report

The revised manuscript took into account the comments.
I recommend publication.

---

## Editorial Decision

published